# Comparing the Size at Onset of Sexual Maturity of Edible Crab (*Cancer pagurus*, Cancridae) in Berwickshire and Northumberland

**Blair Alexander Andrew Easton [1,*], Andrew Boon [2], Joe Richards [3] and Kevin Scott [1]**

[1] St Abbs Marine Station, The Harbour, St. Abbs, Eyemouth TD14 5PW, Scottish Borders, UK; kevin.scott@marinestation.co.uk
[2] Northumberland Inshore Fisheries and Conservation Authority, Blyth NE24 4RT, Northumberland, UK; andrew.boon@nifca.gov.uk
[3] Blue Marine Foundation, London WC2R 1LA, UK; joe@bluemarinefoundation.com
[*] Correspondence: blair.easton@marinestation.co.uk; Tel.: +44-01890-771688

**Abstract:** The literature suggests regional variations in the size at which sexual maturity is reached for commercially important edible crab (*Cancer pagurus*), worth GBP 74.3 million annually, which could have implications for regional fisheries management. Berwickshire and Northumberland are geographically divided by the Scotland and England border and remain within the Berwickshire and North Northumberland Coast SAC (Special Area of Conservation). Each are managed by differing fisheries authorities and Minimum Conservation Reference Sizes (MCRS). Morphometric measurements were recorded for each *C. pagurus* individual to categorise morphometric maturity using segmented regression, with gonadal maturity categorised using visual gonad characteristics and general linear model regressions to compare onset in sexual maturity. Results showed regional variations for gonadal maturity with males reaching sexual onset at a carapace width size of 108.5 mm in Berwickshire and 109.9 mm in Northumberland; females at a size of 126.8 mm in Berwickshire and 120.8 mm in Northumberland. This was also true for morphometric maturity based on chelae height, that males (141.1 mm) and females (134.7 mm) from Berwickshire were morphometrically mature at greater sizes than males (130.1 mm) and females (120.8 mm) from Northumberland. This study shows that the respective MCRS in both regions are appropriate for the *C. pagurus* populations, but implications for fisheries management could be present.

**Keywords:** *Cancer pagurus*; conservation; fisheries management; MCRS; minimum landing size

**Key Contribution:** The size at onset of sexual maturity for *Cancer pagurus* differs between Berwickshire and Northumberland, UK. When the regions are compared, males mature at smaller sizes in Berwickshire (108.5 mm) and females mature at smaller sizes in Northumberland (120.8 mm). These sizes are still appropriate for the current regional minimum conservation reference sizes, but implications for fisheries management could be present.

## 1. Introduction

The edible crab (*Cancer pagurus*, Cancridae) fishery is the most valuable crab fishery in the UK and an important economic product for inshore commercial fishermen. Annual landings were in excess of 31,632 tonnes with a landing value of just under GBP 74.3 million in 2019 [1,2]. The fishery is subject to EU regulations that impose statutory Minimum Conservation Reference Sizes (MCRS) which prohibit the landing and selling of undersized individuals. The MCRS for *C. pagurus* differs between regions and in Scotland the MCRS is 150 mm (carapace width) (except Shetland Isle where it is 140 mm) [3] while Northumberland (specifically the Northumberland Inshore Fisheries and

Conservation Authority), England, MCRS is 130 mm [4]. The primary aim of the MCRS at a management level is to enable individuals to reproduce at least once before harvesting [5].

During 2020, there were 62 active creel fishing vessels (<10 m in length) operating from Eyemouth harbour (55.873726, −2.086900) [6], 75 active vessels in Northumberland, and 11 active vessels (based in Scotland) operating on both sides of the border [2,6]. Landings and effort data are available for 2020; however, COVID-19 restrictions that were in place for 2020 likely affected the fishing effort and landings for this period and therefore are unlikely to adequately represent the region as a whole. In 2020, the *C. pagurus* fishery for Berwickshire was valued at GBP 567,000 and GBP 2.13 million within the Northumberland region, highlighting the economic importance of this fishery to these regions [2,6]. *C. pagurus* landings data for the Berwickshire region and Northumberland region can be seen for 2017–2019 in Table 1.

**Table 1.** Landings and value per annum of *C. pagurus* from 2017 to 2019 for both study regions.

| Region | Landings and Value Per Annum (Tonne)/(£'000) | | |
|:---:|:---:|:---:|:---:|
| | **2017** | **2018** | **2019** |
| Berwickshire | 530/820 | 498/1014 | 463/1081 |
| Northumberland | 1057/1375 | 971/1598 | 949/2140 |

Note: Berwickshire values are taken from the Scottish sea fisheries statistics reports retrieved from https://www.gov.scot/publications/scottish-sea-fisheries-statistics-2018/documents/ (accessed on 5 May 2022). These values are a collation of landings in the main regional port of Eyemouth. Northumberland values were provided by NIFCA datasets.

Increased reports from local fishermen (pers comm), who fish both sides of the Scottish/English border, have suggested that the discrepancy in MCRS between regions, and the close proximity of local fishing fleets to the border, has resulted in cross-border fishing and landings. If this is the case, current MCRS measures between regions are unlikely to be effective as the removal of individuals deemed undersized in one region are landed in the other, further limiting the benefits of maintaining these individuals to provide brood stock. Regional approaches to management are the current norm in managing the edible crab stocks in Berwickshire and Northumberland; however, with reports of cross-border landings, a harmonised MCRS for Berwickshire and Northumberland could prevent such activity. Therefore, understanding the size at which an individual becomes mature in both regions will be beneficial to the fishery and advancement of the fishery management.

*C. pagurus* is a decapod species that has an international distribution spanning from Norway to Morocco, located at depths of up to 100 m [7]. Typically, this species resides in habitats consisting of coarse sediment and rock [7]. The literature suggests that the size at onset of sexual maturity (SOM) of localised populations of *C. pagurus* differs spatially, depending on different environmental factors such as habitat type, temperature, and depth [8,9]. This is also reflected by the variance in MCRSs around the UK coast (Table 2). Maturity can be categorised into four criteria—gonadal, morphometric, behavioural and functional [8]. Each focusses on different stages of the decapods progression through maturity—gonadal is the presence of developed testes for males and ovaries for females; morphometric the changes in growth of chelipeds in males and the abdomen and pleopods for females; behavioural includes the presence of sperm plugs in the females' oviducts as an act of copulation; functional is a combination of the other three indicated by presence of offspring [8]. The hepatopancreas is an essential organ that provides the individual the energy and nutrients required for growth and progressing through reproductive stages; therefore, it provides an index for assessment of health of the individual [7]. Copulation of female *C. pagurus* occurs between December and February when the females have moulted, allowing the males to mate successfully over three to twenty-one days [7]. Oviposition occurs four months post copulation, typically between

January and June, with egg brooding continuing into the next eight months [7]. There have been no comparative studies investigating the SOM (utilising gonadal maturity or sperm plugs) of *C. pagurus* in the Northumberland and Berwickshire regions. It is essential to understand at what stage crabs reach sexual maturity at a regional scale to properly inform sustainable fisheries management and allow *C. pagurus* the opportunity to reproduce at least once before being landed, thus safeguarding and prolonging the fishery for future generations, providing long-term benefits to local, and national economies.

SOM can be used to estimate the reproductive output of individuals before they are caught according to the current MCRS. Here, we conduct a cross-border fisheries project to determine whether the SOM for local *C. pagurus* populations (males and females) in both Berwickshire and Northumberland regions are different and assess whether the current MCRSs are suitable for each region.

**Table 2.** Previous size at onset of sexual maturity (SOM) literature for *C. pagurus* within the UK.

| Region | Sex | Maturity Metric | Size at 50% Maturity ($CW_{50}$) (mm) | Reference |
|---|---|---|---|---|
| North Wales | M | Gonad development | 56–94 | [7] |
| North Wales | F | Gonad development | 86–105 | [7] |
| South Wales | M | Gonad development | 56–94 | [7] |
| South Wales | F | Gonad development | 101–115 | [7] |
| Norfolk England | M | | 105 | [9] |
| Norfolk England | F | | 110 | [9] |
| Selsey England | M | | 115 | [9] |
| Selsey England | F | | 125 | [9] |
| E Coast England | F | Sperm plugs | 116 | [10] |
| SW Ireland | F | Mature gonads | 127–139 | [10] |
| England | M | Chelae | 110 | [10] |
| Shetland Scotland | M | Averaged all | 116 | [11] |
| Shetland Scotland | F | Averaged all | 128 | [11] |
| Shetland Scotland | F | Sperm plugs | 123 | [11] |
| Shetland Scotland | M | Mature gonads | 125 | [11] |
| Shetland Scotland | F | Mature gonads | 133.5 | [11] |
| Shetland Scotland | F | Hatched | 144 | [11] |
| Shetland Scotland | M | Averaged functional | 125 | [11] |
| Shetland Scotland | F | Averaged functional | 139 | [11] |
| Western Channel England | F | Mature gonads | 137–147 | [12] |
| Scotland East and West | M | Mature gonads | 101–106 | [13] |
| Scotland East and West | F | Mature gonads | 127–128 | [13] |
| Scotland | F | Sperm plug | 122.9 | [14] |
| Scotland | F | Ovigerous | 143.7 | [14] |
| Ireland | F | Mature gonads | 132–138 | [15] |
| Eastern Channel England | M | Mature gonads | 105 | [16] |
| Eastern Channel England | F | Mature gonads | 126 | [16] |
| Western Channel England | M | Mature gonads | 90 | [16] |
| Western Channel England | F | Mature gonads | 112 | [16] |
| North Sea | M | Mature gonads | 89 | [16] |
| North Sea | F | Mature gonads | 109 | [16] |
| Ireland | F | Gonad development | 120 | [17] |
| Isle of Man, Irish Sea | M | Gonad maturity | 89 | [18] |
| Isle of Man, Irish Sea | F | Gonad maturity | 108 | [18] |

Note: $CW_{50}$ represents the carapace width (mm) at which 50% of the population are deemed sexually mature.

## 2. Materials and Methods

*C. pagurus* individuals were collected from the Berwickshire region by local inshore fishermen over three fishing trips. Derogations were granted by Marine Scotland over the sampling period to allow the landing of individuals below the Scottish MCRS of 150 mm. The surveys were in accordance to terms of Section 9 of the Sea Fish Conservation Act 1967, Article 25 of Council Regulation No. 2019/1241, the specified crustaceans (prohibition on landing, sale and carriage) (Scotland): order 2017 No. 455 and the undersized edible crabs (Scotland) order 200 No. 228. In the Northumberland region (River Tyne to the Scotland/England Border) inshore creel fleets (eight fishing trips) deployed by the Northumberland Inshore Fisheries Conservation Authority (NIFCA) and local fishing vessels in Northumberland (two fishing trips) were used for sample collection (Figure 1).

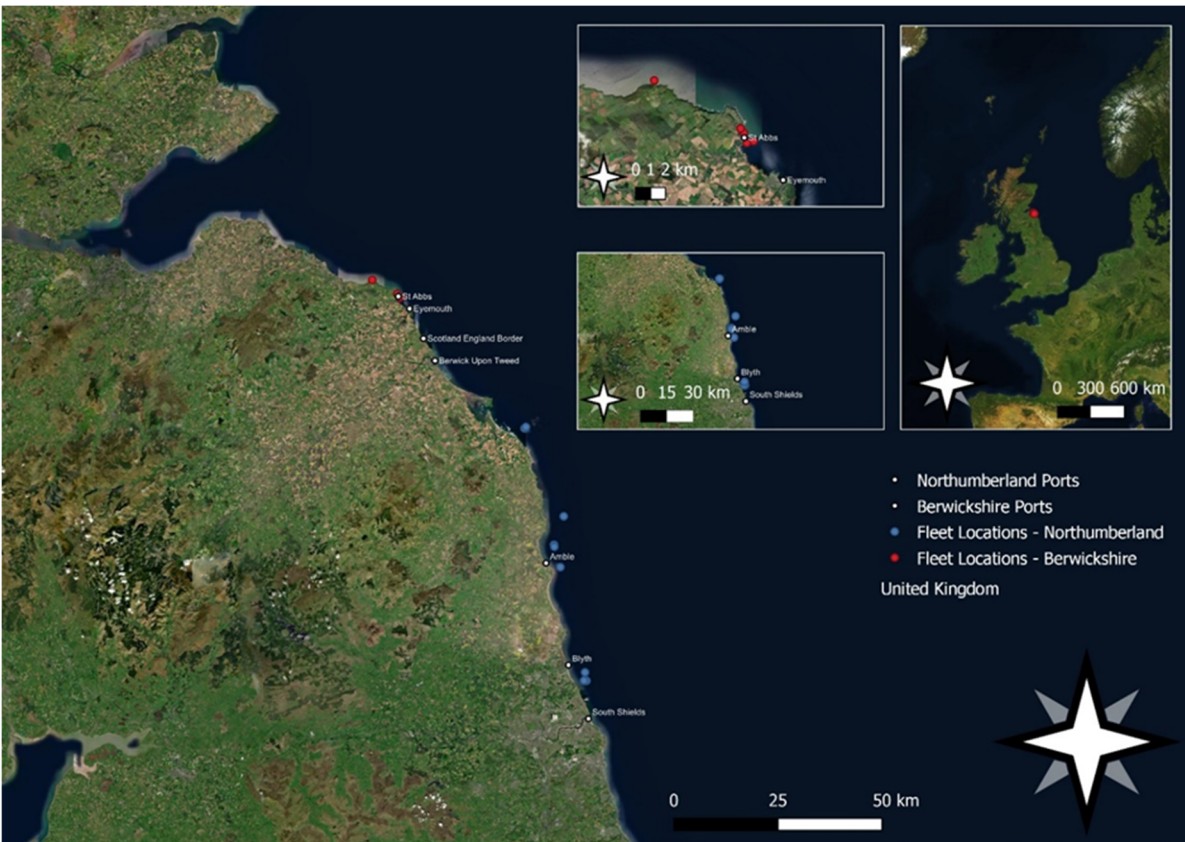

**Figure 1.** Map of locations from which the *C. pagurus* were sampled from both the Berwickshire (55.902769, −2.128988) and Northumberland (55.137154, −1.437651) areas of survey. Sampling locations for Northumberland are highlighted by the blue dots whereas the sample locations for Berwickshire are highlighted by the red dots.

A total of 768 individuals (carapace width range: 60–209 mm) were collected from both sites between September 2020 and May 2021 which were processed at St. Abbs Marine Station. Those *C. pagurus* individuals collected from the Berwickshire region were held in 3000–10,000 L aquarium tanks filled with ambient seawater (mean ± sem: 12.25 ± 0.46 °C, 33.84 ± 0.13 ppt). Those collected from the Northumberland district were frozen prior to arrival to the marine station due to storage and transportation difficulties. These samples were defrosted over a 24 h period at room temperature (22 °C) to allow complete thawing of the organ tissue. Thawing of the organ tissue allowed for the organs and tissue to be removed more easily, limiting the chance of wasted target tissue.

Initially, each fresh individual from Berwickshire was subject to a cold shock treatment, whereby the individual was placed in a freezer for a duration of 30 to 60 min

at a temperature range of −16 to −20 °C. This treatment allowed the individuals to be desensitised before the dissections took place. To ensure the individual was dead before dissection, it was monitored for 5 min during which time any movement from chelae, pleopods and eye stalks was associated with inadequate desensitising and the individual was placed back in the freezer for an additional 30 to 60 min. Before each dissection took place, morphological data were collected from each individual using measurement callipers (Proops 350 mm Inside Outside Plastic Calliper Metric Measuring Scale) and a calibrated AMIR™ digital scale. For each individual, the following were recorded: wet weight (g), carapace width (mm), chelae length (mm), chelae height (mm), chelae depth (mm), sex, condition index, and moult stage. Morphological measurements of the chelae were assigned to the right claw. If the individual had a regrowing right chela, or had the right chela missing, the left chela was used for these measurements as the chelae are not dimorphic [7,19]. The condition index is a method of determining the general condition of an individual's morphometry based on criteria used in [7] (Table 3). The extent of black spot coverage on the body was also noted for each individual based on criteria from [7,10,20]. The moult stage was determined using the same table of criteria used in the paper [7] (Table 4).

**Table 3.** Descriptions detailing the criteria to meet each condition index stage.

| Condition Index | Condition Description |
|---|---|
| 1 | High standard of health. Chelipeds present. Chelae present. No black spot lesions. No damage. |
| 2 | Good standard of health. One to two chelipeds are missing. Limited black spot lesions. |
| 3 | Average standard of health. More than two chelipeds missing. Black spot lesions present. |
| 4 | Bad standard of health. One or both chelae missing. Black spot lesions covering around 50% of the carapace. Limited damage to the body. |
| 5 | Poor standard of health. One or both chelae missing and chelipeds missing. Large surface area of black spot lesions on the body. Damaged carapace. |

Note: The information in this table is developed from [7,10,20].

**Table 4.** Visual descriptions of moult stages to assign to each individual *C. pagurus*.

| Moult Stage | Description |
|---|---|
| Early Post Moult | Soft, white, no biofouling on carapace, very sharp toes. |
| Recent Moult | No biofouling on carapace, sharp toes, carapace not fully hardened. |
| Inter-Moult | Carapace covered with biofouling usually in the form of biofilm attached to the hairs of the chelipeds, toes are worn to a smooth rounded shape. |
| Degraded | Carapace shows signs of great biofouling in the form of tubeworms, barnacles, large surface area coverage of biofilms especially attached to the hairs of the chelipeds, damage to carapace (holes, indentations), distinguishable smell to the individual. |

Note: Details and descriptions used to assign the stages were based on those from [7,21].

Females were observed for sperm plugs prior to dissection, which were noted if present. The crab dissection took place on the ventral side of the animal. An incision at the telson allowed for the dissection to take place along the moult line. This allowed the dorsal and ventral sides to be separated, opening the body cavity. Once the body cavity was exposed, the hepatopancreas, which sits on top of the gonads, was removed for weighing post dissection. The gonads were then exposed and a photo was taken for post sampling analysis. The sex stage was then determined by comparing the photo with information and details from the literature [7,10,19,20]. An example of the information and details is summarised in Table 5, Figure 2.

**Table 5.** Visual descriptions used to assign the gonad stages for each *C. pagurus* individual.

| Female | | | | | |
|---|---|---|---|---|---|
| Stage | 1-Immature | 2-Undeveloped | 3-Developing | 4-Mature | 5-Recovery |
| Details | No egg cells present | Pre-vitellogenesis | Early secondary vitellogenesis | Late secondary vitellogenesis | Post Reproductive |
| Visual | Gonad is transparent and thin in structure. | Gonad lobes are visible with a light pink/grey coloration. | Gonad has more noticeable pink colour. Covers less than 50% of the cavity. | Ovaries are very large, covers over 50% of the cavity with a prominent orange/red colour. | Remnant eggs are visible with the ovary exhibiting a loose structure and white appearance. |
| Male | | | | | |
| Stage | 1-Immature | 2-Developing | 3-Mature | | |
| Details | Spermatids | Spermatozoa | Spermatophore | | |
| Visual | Small testes that are transparent or undetectable. | White and obvious testes. | Both swollen testes and vas deferens. | | |

Note: Descriptions are based on those by [7,8,10,19,21].

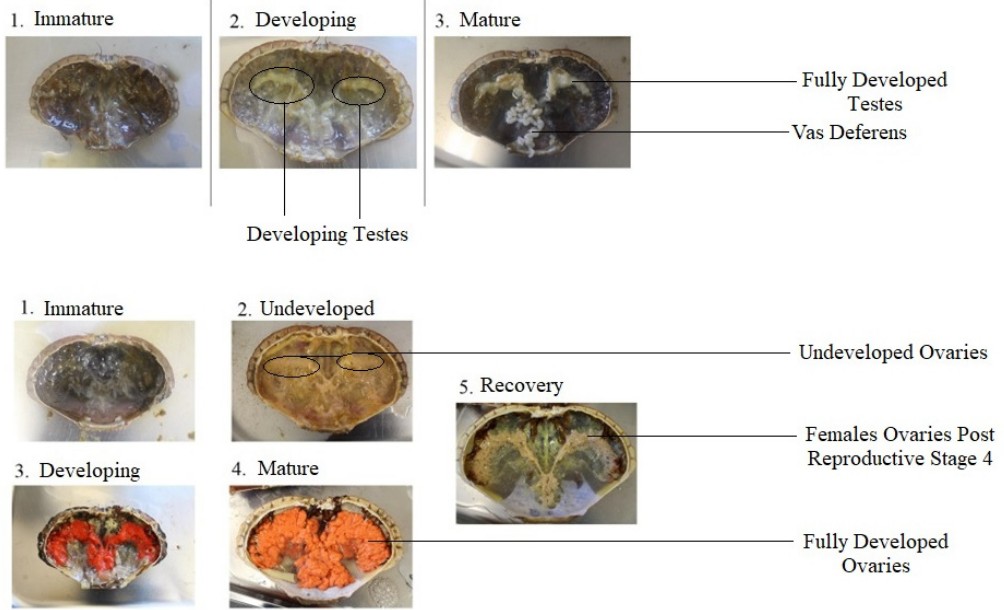

**Figure 2.** Visual representations of the three gonad stages of male *C. pagurus* and five gonad stages of female *C. pagurus*. For males, stages 1 to 3 are presented from left to right at the top of the image. For females, stages 1 and 2 are presented left to right on the top, stages 3 and 4 are presented left to right on the bottom. Stage 5 is shown on the far right.

The wet weight of the hepatopancreas was measured using a calibrated Sartorius™ AC 211S-00MS Iso Cal digital scale. If the gonad stage for males were three, and three or four for females, deeming the individual sexually mature, the gonad was removed and the wet weight recorded to the same criteria as the hepatopancreas. The hepatosomatic index (HSI), a means of indicating lipid stores in the individual, is calculated by the hepatopancreas wet weight (HWW) (g) divided by the total wet weight (WW) (g) of the individual to give a percentage value (HSI).

$$HSI = HWW/WW \times 100$$

Similar to the hepatosomatic index, the gonadosomatic index (GSI) is calculated by the gonad wet weight (GWW) (g) divided by the total wet weight (WW) (g) to provide a percentage value.

$$GSI = GWW/WW \times 100$$

### 2.1. Statistical Analysis

All statistical analysis was performed in R Studio [22]. Initial data were analysed for normal distribution and homogeneity of variance using the Shapiro–Wilk test, normality histograms, and Levene's Test. Normality and homogeneity of variance was considered when significance was interpreted as *p*-value > 0.05. Size at gonadal and morphometric maturity of the individuals was estimated using the sizeMat package (version 1.1.2, published: 2 June 2020) [23]. Segmented regression analysis and models were conducted using the segmented package (version 1.3.4, published: 22 April 2021) in R Studio. Data were separated by location (Berwickshire and Northumberland) to allow for regional comparisons. Comparisons made regarding the hepatosomatic index (HSI) were analysed using Anova tests partnered with post hoc Tukey HSD tests. The gonadosomatic index (GSI) was tested with sex, gonad stage, and condition index using linear regressions. All statistics were tested to the significance value of 0.05.

### 2.2. Morphometric Maturity

Individuals were assigned into two groups (immature = 0 and mature = 1) which were classed using the allometric measurements (frequentist logit approach), X = independent variable (carapace width) and Y= dependent variable (chelae height, depth, and length) [24]. $CW_{50}$ is the value given when there is a 50% chance at a given carapace width (mm) the individual is considered mature [25,26]. In the regression analysis, X is considered the explanatory variable, in this study carapace width (mm), and the classification of maturity CS (juveniles: 0, adults: 1) is considered the response variable (binomial). The variables are fitted to a logit function [24]:

$$PCS = 11 + e^{-(\beta^0 + \beta^1 x)}$$

where *PCS* is the likelihood of an individual being mature at carapace width (x). $\beta^0$ (intercept) and $\beta^1$ (slope) are parameters estimated. The $CW_{50}$ is calculated as:

$$CW_{50} = -\beta^0 \beta^1$$

Maturity ogives were presented as graphs which highlighted the $CW_{50}$ for each location and sex. Segmented regression was applied to assess breakpoints (BP) and confidence intervals at which morphometric data collected (chelae depth, chelae height and chelae length) indicated morphometric maturity in relation to allometric relationships [23]. The segmented regression follows the process of measuring the distance between two fitted lines at each respective breakpoint using the minimisation of a parameter [23].

### 2.3. Physiological Maturity

The size at physiological maturity is assessed using the carapace width (mm) in relation to the maturity stage assigned during the histological dissection process. The function follows a logistic approach in which the logit regression is based on a general linear model (GLM) [23]. As per the morphometric maturity, the function requires an allometric variable (X), in this case carapace width (mm) and stage of sexual maturity (immature = 0, mature = 1). Similar to the morphometric ogives, the gonad maturity ogives were presented as the fitted values as a curve logistic regression with confidence intervals (95%). Also highlighted is the $CW_{50}$ for each location and sex.

## 3. Results

From both regions of sampling (Berwickshire and Northumberland), 768 individual *C. pagurus* were collected and dissected between the period of September 2020 and May 2021. A total of 501 males and 267 females were collected with 283 (56.49%) of these individuals considered mature (gonad stage three for males; three and four for females). Across the sampled population used in this study, 78.45% (*n* = 222) of the cohort were categorised as mature from the Berwickshire group and 21.55% (*n* = 61) from the Northumberland group. The smallest carapace width (mm) recorded was 60 mm and the largest being 209 mm (Figure 3 . The smallest recorded wet weight (g) was 25.2 g and largest being 1345 g (Figure 4). Location-specific results showed that of those collected in Berwickshire, 222 were considered mature (190 males, 32 females) out of 437 individuals (313 male, 124 female). Morphological measurements showed a carapace width range of 72–209 mm and wet weight range of 50–1345 g. Those collected in Northumberland, 61 were considered mature (56 males, 5 females) out of 332 individuals (88 males, 144 females). Morphological measurements showed a carapace width range of 60–186 mm and a wet weight range of 25.2–810.6 g. All females were assessed for the presence of sperm plugs, nine females in Berwickshire and 25 females in Northumberland presented sperm plugs. Black spot was also recorded as a measure of condition for the health of the individuals sampled. A total of 31 individuals in Berwickshire and 11 individuals from Northumberland were shown to have black spot present. Out of all individuals collected and sampled, 76 had their left claw measured due to missing or re-growing right chela. Of these 76 individuals, 49 were collected from Northumberland (males = 32, females = 17) and 27 from Berwickshire (males = 19, females = 8). Condition of the individuals did not greatly differ between the regions of sampling, those sampled from Berwickshire; males averaged 2.14 ± 0.06 and females 2.21 ± 0.10. Those sampled from Northumberland showed the greater condition indices on average, with males 3.17 ± 0.08 and females 2.78 ± 0.10, showing worse conditioned individuals. A full summary of the measurements taken is presented in Table 6.

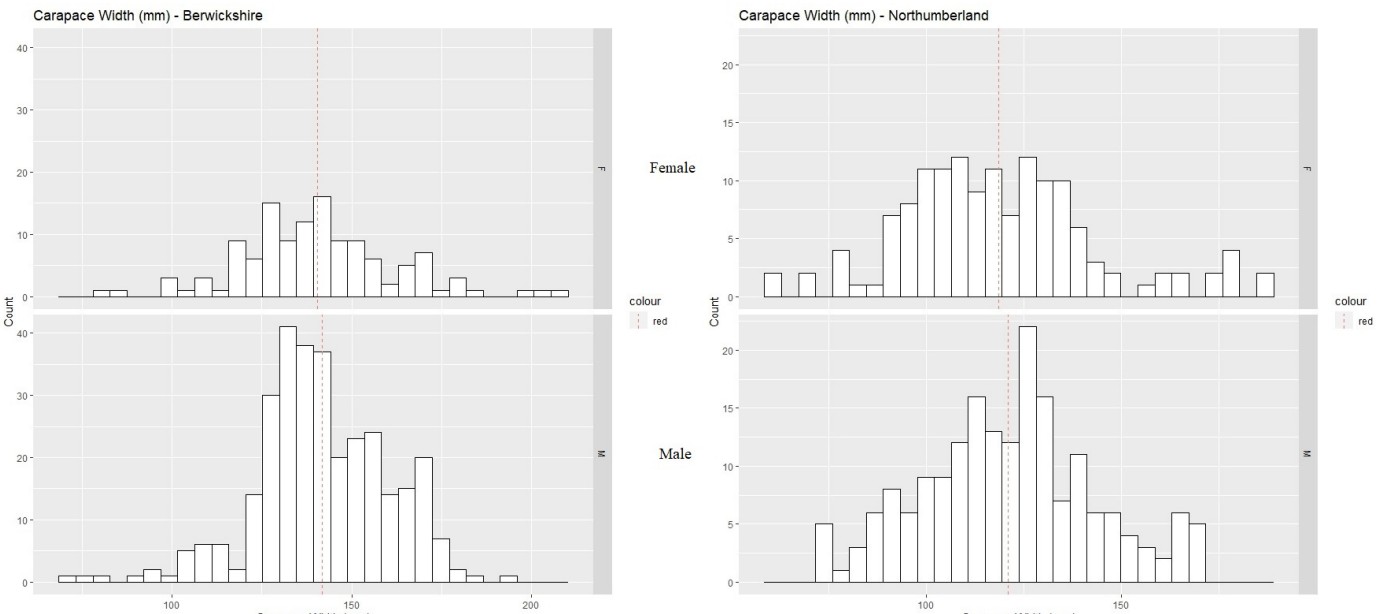

**Figure 3.** Carapace width (mm) of the Berwickshire (*n* = 437; (Males = 313, Females = 124)) and Northumberland (*n* = 332; (Males = 188, Females = 144)) *C. pagurus* sampled cohorts. The red dotted line highlights the mean carapace width (mm) for each population separated by sex.

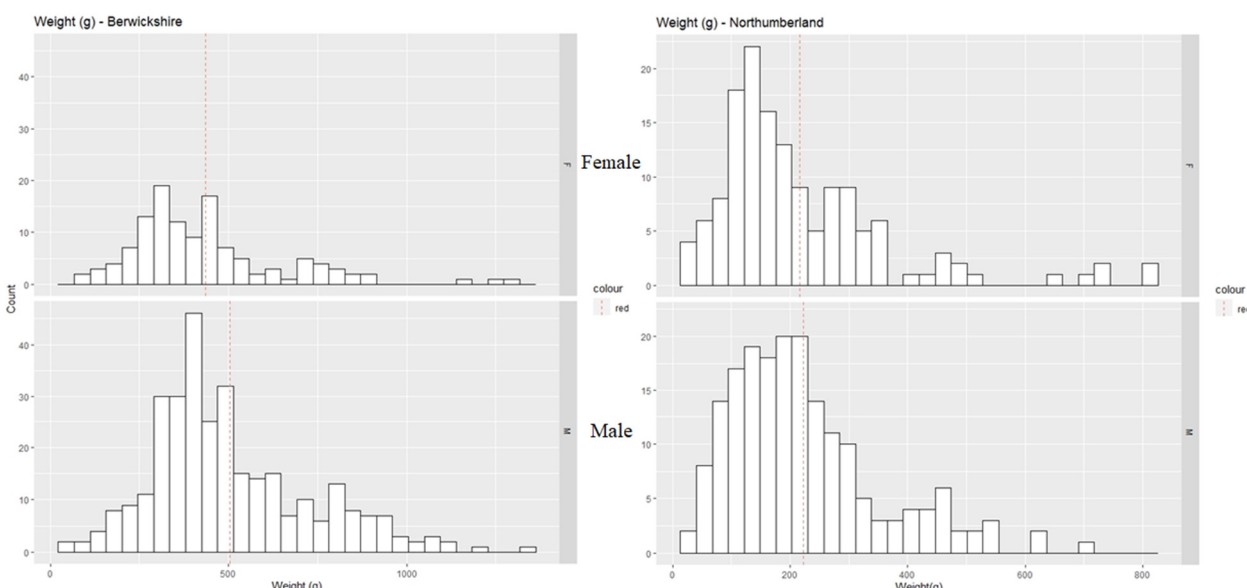

**Figure 4.** Wet weight (g) of the Berwickshire ((*n* = 437); (Males = 313, Females = 124)) and Northumberland ((*n* = 332); Males = 188, Females = 144) *C. pagurus* sampled cohorts. The red dotted line highlights the mean wet weight (g) for each population separated by sex.

**Table 6.** Mean and standard error of the morphometric measurements taken during the pre-dissection stage of each *C. pagurus* individual from both regions.

| Region | Sex | CW (mm) | WW (g) | CI | MS | CL (mm) | CD (mm) | CH (mm) | HW(g) | GS |
|---|---|---|---|---|---|---|---|---|---|---|
| Berwickshire | M | 141.76 ± 1.05 | 503.52 ± 12.60 | 2.14 ± 0.06 | 3.16 ± 0.03 | 34.51 ± 0.39 | 23.22 ± 0.29 | 37.18 ± 0.43 | 33.44 ± 0.77 | 2.45 ± 0.04 |
| | F | 140.48 ± 1.99 | 436.33 ± 19.93 | 2.21 ± 0.10 | 3.11 ± 0.07 | 27.57 ± 0.46 | 18.18 ± 0.29 | 29.50 ± 0.46 | 69.55 ± 34.72 | 2.29 ± 0.11 |
| Northumberland | M | 121.03 ± 1.63 | 222.89 ± 9.40 | 3.17 ± 0.09 | 2.89 ± 0.05 | 26.75 ± 0.55 | 17.55 ± 0.40 | 29.04 ± 0.63 | 11.65 ± 0.70 | 1.99 ± 0.06 |
| | F | 118.56 ± 2.07 | 216.34 ± 12.65 | 2.78 ± 0.11 | 2.70 ± 0.06 | 23.20 ± 0.46 | 14.77 ± 0.30 | 24.83 ± 0.48 | 11.59 ± 0.90 | 1.93 ± 0.12 |

Note: All data are presented as mean ± sem. Abbreviations in the table are as follows: CW—carapace width, WW—wet weight, CI—condition index, MS—moult stage, CL—chelae length, CD—chelae depth, CH—chelae height, HW—hepatopancreas weight, and GS—gonad stage.

### 3.1. Physiological Maturity

Physiological maturity ($CW_{50}$) was considered using the maturity stages three for males and three–four for females. Following the logistic regression with bootstrapping, it was considered that the size at which gonadal maturity was met for 50% of the population to be 108.5 mm (95% CI, 97.7–116.4 mm) for males in Berwickshire and 109.9 mm (95% CI, 105.9–113.5 mm) for those in Northumberland (Figure 5. For females, this was 126.8 mm (95% CI, 122.2–130.9 mm) in Berwickshire and 120.8 mm (95% CI, 117.2–125.3 mm) in Northumberland (Figure 5). In both regions, the females showed maturity at larger carapace widths when compared to males. Considering the ogive results, the *C. pagurus* in the Northumberland district would be mature at the current MCRS of 130 mm, and the same for the *C. pagurus* cohort from Berwickshire under the current Scottish MCRS of 150 mm.

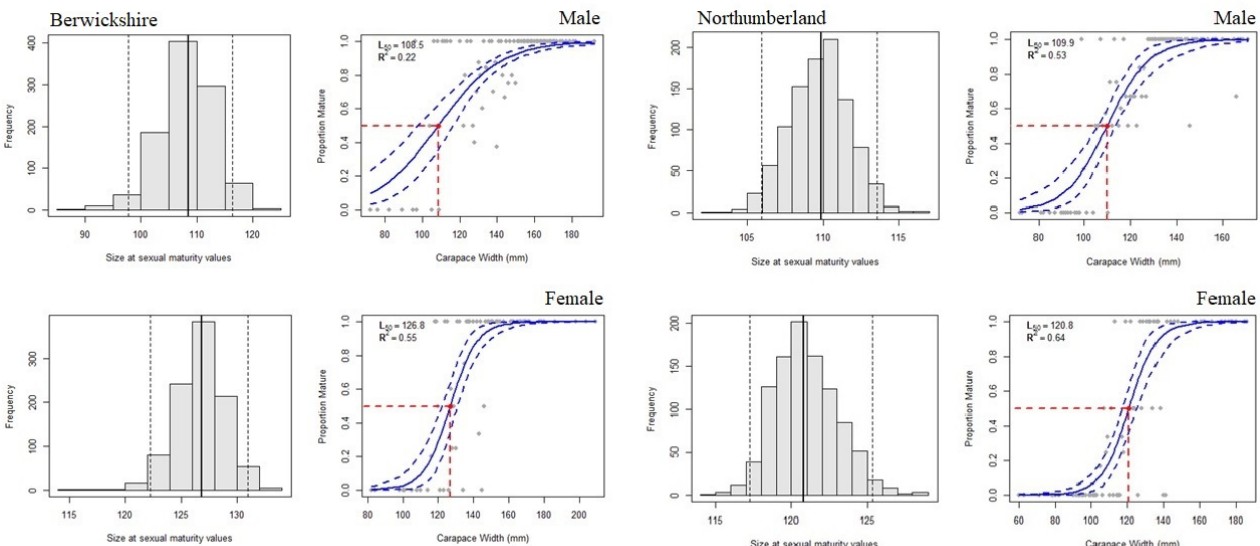

**Figure 5.** Male and female gonad maturity of the *C. pagurus* cohorts sampled from Berwickshire and Northumberland. The point at which 50% of the population are said to be mature is highlighted in red ($CW_{50}$), with confidence intervals (95% CI) shown by the blue dashed lines.

### 3.2. Morphometric Maturity

Segmented regression was used to calculate the carapace width (mm) in which there were changes in allometry relationships as described in [23]. The parameters used chelae length (mm), chelae depth (mm), and chelae height (mm) in regard to male *C. pagurus*. Outputs from the segmented regression for the male individuals in Berwickshire showed the chelae length break point (BP) at 119.07 ± 12.37 mm (mean ± sem), chelae depth breakpoint (BP) at 83.99 ± 7.33 mm (mean ± sem), and chelae height breakpoint (BP) at 137.99 ± 9.77 mm (mean ± sem). Outputs from the segmented regression for the male individuals in Northumberland showed the chelae length break point (BP) at 129 ± 6.72 mm (mean ± sem), chelae depth breakpoint (BP) at 119.67 ± 5.68 mm (mean ± sem), and chelae height breakpoint (BP) at 129 ± 4.77 mm (mean ± sem).

Using the sizeMat package in R Studio to determine the morphometric maturity of the *C. pagurus* individuals sampled in both regions, the relationship (linear regression) between carapace width (mm) and chelae height (mm) was tested. This method allowed for morphometric maturity analysis of females. In Berwickshire, the carapace width at which morphometric maturity was met for males was 141.1 mm ($R^2$ = 0.7, CI = 139.4–143 mm) and 134.7 mm ($R^2$ = 0.89, CI = 132.8–136.8 mm) for females. In Northumberland, carapace width at which morphometric maturity was met was 130.1 mm ($R^2$ = 0.84, CI = 128–132.5 mm) for males and 120.8 mm ($R^2$ = 0.64, CI = 117.2–125.3 mm) for females. A summary of the results from both regression analyses used can be seen in Table 7.

**Table 7.** Summary of outputs from the regression analysis (segmented and linear) considering the relationship between carapace width (mm) and other morphometric measurements (chelae height (CH), chelae depth (CD) and chelae length (CL) (mm)) from sampled individuals of *C. pagurus* from Berwickshire and Northumberland.

| Region | Measurement (mm) | Sex | N | Segmented Regression | | | Linear Regression | | |
|---|---|---|---|---|---|---|---|---|---|
| | | | | Slopes | BP (mm) | $R^2$ | $R^2$ | BP (mm) | CI |
| Berwickshire | CH | M | 308 | 0.187, 0.233 | 137.99 ± 9.77 | 0.85 | 0.7 | 141.1 | 139.4–143 |
| | CD | M | 308 | 0.108, 0.125 | 83.99 ± 7.33 | 0.76 | 0.83 | 142.3 | 140.9–143.7 |

| | | | | | | | | |
|---|---|---|---|---|---|---|---|---|
| | CL | M | 308 | 0.166, 0.225 | 119.06 ± 12.37 | 0.86 | 0.86 | 153.3 | 151.8–154.9 |
| | CH | F | | | | | 0.89 | 134.7 | 132.8–136.8 |
| Northumberland | CH | M | 170 | 0.178, 0.266 | 129 ± 4.77 | 0.92 | 0.84 | 130.1 | 128–132.5 |
| | CD | M | 170 | 0.105, 0.155 | 119.67 ± 5.68 | 0.89 | 0.92 | 121.8 | 120.4–123.3 |
| | CL | M | 170 | 0.012, 0.015 | 129 ± 6.72 | 0.89 | 0.93 | 121.2 | 119.9–122.5 |
| | CH | F | | | | | 0.64 | 120.8 | 117.2–125.3 |

Note: The segmented regression is associated with the morphometric maturity and the linear regression is associated with the gonad maturity. BP is the estimated carapace width breakpoint.

### 3.3. Hepatopancreas Weight and Hepatosomatic Index (HSI)

In Berwickshire (Figure 6), the hepatopancreas wet weight was significantly different dependent on the gonad maturity stage of both sexes (F-value = 5.59, $p < 0.05$), in which gonad maturity stages two, three and four ($p < 0.05$) were different to gonad stage one. Hepatopancreas weight for males with gonad stages one, two and three ($p < 0.05$) was significantly different to females of gonad stage two. In comparison, for both sexes in Northumberland (F-value = 31.82, $p < 0.05$) (Figure 6) their hepatopancreas wet weight was significantly different between gonad stage one and all other sex-specific gonad stages ($p < 0.05$). Gonad stage four in females was significantly different to gonad stages two, three and five ($p < 0.05$). Variations in hepatopancreas weight were found between the sexes at different *C. pagurus* gonad stages in Northumberland. Gonad stage one in males was significantly different to gonad stage two in females ($p = < 0.05$). Male gonad stages one, two and three ($p < 0.05$) were significantly different to female gonad stage four. Only gonad stages one ($p < 0.05$) and two ($p = 0.01$) in males showed significant difference from female gonad stage five.

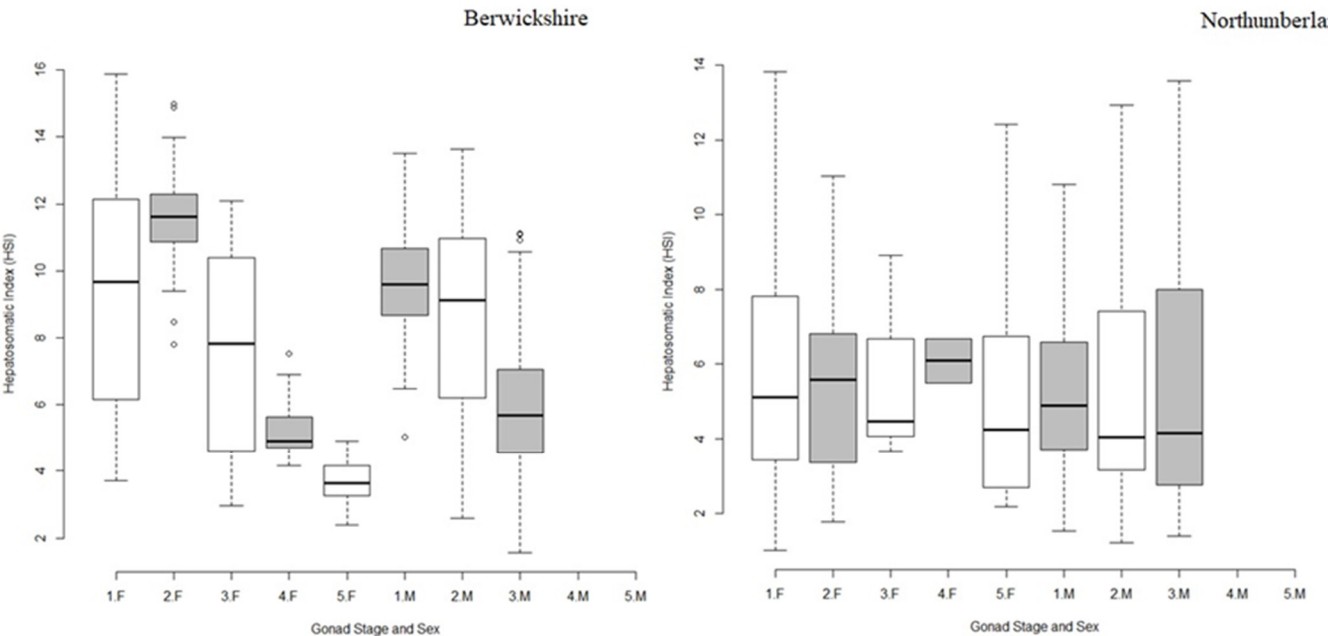

**Figure 6.** Hepatosomatic index (HSI) of both sexes considering all gonad maturity stages from individuals sampled in Berwickshire and Northumberland. Data are presented as mean ± sem. Circles indicated in the image present outliers post data analysis.

The HSI values for the Berwickshire-sampled individuals showed significant difference by gonad maturity stage (F-value = 61.83, $p < 0.05$), sex (F-value = 23.93, $p < 0.05$), and gonad stage as a covariate with sex (F-value = 11.78, $p < 0.05$). Contrastingly, the HSI for the individuals sampled from Northumberland showed no significance for gonad

stage (F-value = 0.25, $p$ = 0.90), sex (F-value = 0.78, $p$ = > 0.05), or as covariates (F-value = 0.02, $p$ = 0.97). Further results indicated that the HSI of the gonad stages one and two are significantly different to all other gonad stages ($p < 0.05$). There was a significant difference between gonad stages five and three ($p$ = 0.01). Differences found between covariates indicated that gonad stages one and two in males showed significant difference to the female gonad stages two, four, and five ($p < 0.05$). Male gonad stage three showed significant difference from female gonad stage two ($p < 0.05$). The condition of the individual was shown to affect the HSI of the individuals (Anova, F-value = 5.80, $p < 0.05$), in particular that condition stage three was significantly different to stages one and two (Tukey HSD, $p < 0.05$). As covariates, condition and sex did not seem to affect the HSI value (Anova, F-value = 1.12, $p$ = 0.34). This considers those sampled from Berwickshire (Figure 7), as for those in Northumberland (Figure 7) the opposite occurred, as covariates condition and sex affected the HSI significantly (Anova, F-value = 6.00, $p < 0.05$). Condition stage one in males was significantly different for condition stages four and five (Tukey HSD, $p < 0.05$). Condition stage four was significantly different to stage two (Tukey HSD, $p < 0.05$) for males. The HSI values were also significantly different between the first condition stages in males and females in Northumberland (Tukey HSD, $p$ = 0.02). As separate factors, condition (Anova, F-value = 1.62, $p$ = 0.16) and sex (Anova, F-value = 0.32, $p$ = 0.57) did not affect HSI in Northumberland.

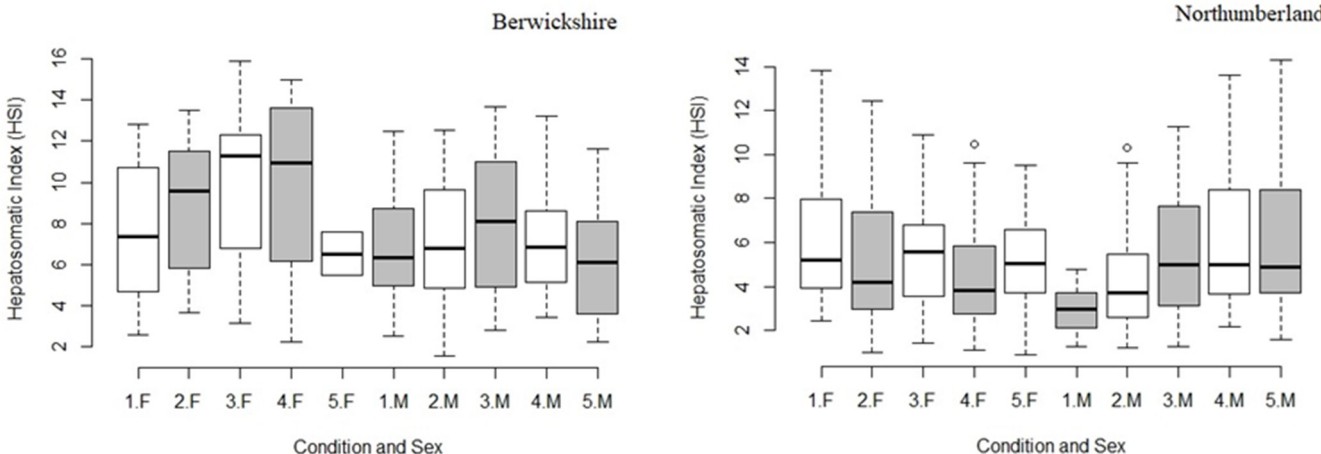

**Figure 7.** Hepatosomatic index (HSI) based on the condition and sex of the individuals sampled from Berwickshire and Northumberland. Data are presented as the mean ± sem. Circles indicated in the image present outliers post data analysis.

### 3.4. Gonad Weight and Gonadosomatic Index (GSI)

The GSI values for the sampled individuals were separated by region of sampling. For those sampled from Berwickshire, the GSI was significantly different regarding sex (F-value–56.76, $p < 0.05$) and gonad stage (F-value = 40.99, $p < 0.05$) following linear regression ($R^2$ = 0.42, $p < 0.05$). For those individuals from Northumberland, the GSI was significantly different for sex (F-value = 25.74, $p < 0.05$), gonad stage (F-value = 3.38, $p$ = 0.04), and condition index (F-value = 5.23, $p$ = 0.02), following linear regression ($R^2$ = 0.35, $p < 0.05$). The GSI for males showed no great difference between the condition indices in Berwickshire, but showed a relatively incremental change with condition indices in Northumberland (Table 8). The GSI of the females showed the greatest value during condition index two in both regions (Table 8).

**Table 8.** The gonadosomatic index for each condition index based on sex of the individuals.

| Sex | Condition Index | Gonadosomatic Index (GSI) | |
|---|---|---|---|
| | | Berwickshire | Northumberland |
| M | 1 | 2.32 ± 0.09 | 1.60 ± 0.51 |
| | 2 | 2.15 ± 0.10 | 1.28 ± 0.16 |
| | 3 | 1.95 ± 0.21 | 2.07 ± 0.37 |
| | 4 | 2.80 ± 0.21 | 2.61 ± 0.31 |
| | 5 | 2.87 ± 0.19 | 2.65 ± 0.30 |
| F | 1 | 4.86 ± 0.87 | 5.11 ± 1.86 |
| | 2 | 7.39 ± 1.29 | 6.03 ± 3.05 |
| | 3 | 1.37 ± 0.41 | 2.34 |
| | 4 | 4.06 ± 0.83 | NA |
| | 5 | 4.40 | NA |

Note: The table presents the gonadosomatic index for the sampled individuals from Berwickshire and Northumberland as mean ± sem. No individuals collected from Northumberland were categorised as condition indexes 4 and 5. Only one female individual was category 5 from the Berwickshire cohort and category 3 from Northumberland. NA indicated in the table highlights non applicable values as no sampled female individuals (n = 0) from Northumberland were considered condition index value 4 or 5.

## 4. Discussion

### 4.1. Physiological Maturity

The size at onset of sexual maturity based on the gonad characteristics used in this study highlights a difference between the study regions of Berwickshire and Northumberland (Figures 8 and 9). Across the sampled population, 78.45% of the cohort were categorised as physiologically mature from the Berwickshire group and 21.55% from the Northumberland group. Previous studies have shown that *C. pagurus* with carapace widths of <100 mm can equate to 38% of the sampled cohort of which 25% females and 50% of males were deemed physiologically mature; this was not found for individuals in Berwickshire or Northumberland [7,26]. Males showed a 1.4 mm $CW_{50}$ difference between Berwickshire and Northumberland, with those exhibiting a greater $CW_{50}$ in Northumberland. Males are considered to reach sexual maturity at smaller sizes than females [7,27]. Contrastingly, females in Northumberland expressed a lower $CW_{50}$ than those in Berwickshire with a $CW_{50}$ difference of 6 mm between the regions. The ability for males to mature at smaller sizes could be of benefit by improving the probability of mating success in populations where the ratio is in favour of females and competition is higher. Discrepancies between regions in relation to sexual maturity is common, as previous studies have shown that males can express a $CW_{50}$ range of 56–125 mm and females 86–133.5 mm across the UK, and in a recent study by [8] the $CW_{50}$ for East Scotland was 101–106 mm for males and 127–128 mm for females which coincides with the results found in the literature [8,11,28]. Previous reports from stock assessment surveys by NIFCA suggested a $CW_{50}$ of 89.5 mm for males and 111.6 mm for females [29]. The female $CW_{50}$ value from this assessment follows suit to the value of [28] at 112 m; however, from this study there is a 8–14 mm difference when both locations are considered which suggests a regional variation across a latitudinal range. It should be stated that the study by [29] used parameters specified by CEFAS (The Centre for Environment, Fisheries and Aquaculture Science), which are used for their assessments, to inform stock assessments rather than specific SOM. In this study, an increase of 20.4 mm for males and 9.4 mm for females over the three-year period was reported (2019–2021). This increment in $CW_{50}$ could be due to the smaller sample size (*n* = 332) compared to the stock assessment by NIFCA (*n* = 11811) [29]. The current MCRS values for these regions (150 mm for Scotland and 130 mm for England) are appropriate for each respective region, allowing 50% of the populations to reproduce before the probability of being

harvested. It has been suggested that the regional variations could be due to the availability of sexual partners, population density, and environmental factors [7,8,10,12,19,30–32]. Environmental factors such as temperature and chemicals have been suggested as a determinate of ovary maturation, with warmer waters associated with lower SOM and chemicals associated with embryonic mortality and lower egg production [32–34]. Growth rates vary after the puberty moult whereby the females express more energy into reproduction than the males [31,9]. It is suggested that static gear contain disproportionately greater numbers of larger crabs than the use of trawling and other active mobile methods of fishing [7]. The period of sampling covered the months of September 2020 and April/May 2021. It has been suggested that the spawning period of *C. pagurus* is in the winter months [13] and more specifically between November and January [12,29], suggesting that the cohort selected for this study would have been in their final stages of reproduction. From all 768 individuals sampled, only two females bore eggs. Thus, no functional maturity conclusions could be suggested in this study. Ovigerous females are rare to catch using commercial fishing methods [19,35,36] which has been suggested due to nesting behaviours and burying in finer sediment offshore. It has been observed that the typical carapace width range for females bearing eggs is between 113 mm and 144 mm in Northern Europe with a minimum size of 129 mm associated with samples in Northern England [21] and with increasing size comes greater fecundity in this species [37,14]. Associating this minimum size from [37] to the female $CW_{50}$ values in this study, it would be suggested there are mitigative measures to allow breeding before removal from the fishery. However, this only considers 50% of the population and there may be individuals removed at larger sizes bearing eggs which are regarded as the minimum size for the stock. Females were observed for presence of sperm plugs to which a total of 34 individuals (Berwickshire: 9; Northumberland: 25) presented one or two sperm plugs. The size of females presenting sperm plugs ranged from 112 to 186 mm in Northumberland and 118 to 170 mm in Berwickshire which is lower than the resultant ~80 mm shown in previous studies [8,17]. It is suggested that the presence of behavioural maturity is met at lower carapace widths than suggested from the gonadal maturity results in this study [8,17]. Females are considered fully mature when gonadal and morphometric maturity are met [8,17], such as the production and carrying of eggs [8,17,34]. Males are considered mature when copulation is successful [21]; however, in this study six individuals (Berwickshire: four (CW: 129–174 mm); Northumberland: two (CW: 129–166 mm)) only presented one teste (carapace width: 129–174 mm). It is unknown whether this affects the success rate for copulation in these males. The interpretation of these results should be that this is not representative of the whole *C. pagurus* population sampled from Northumberland and Berwickshire. Using commercial vessels which were utilised to collect undersized individuals primarily with NIFCA deployed creels collecting those from a broader size range. It could be suggested that this would skew the data, therefore influencing the percentage of those deemed mature. However, the size range as presented in Figures 3 and 4 shows that the size distributions in each region (Berwickshire: 72–209 mm; Northumberland: 60–186 mm) did not vary heavily and therefore could be said that the results are representative, which is supported further by [38] who sampled a wide size range of *C. pagurus* representative of the commercial fishery but also suggested that this be considered when interpreting the results.

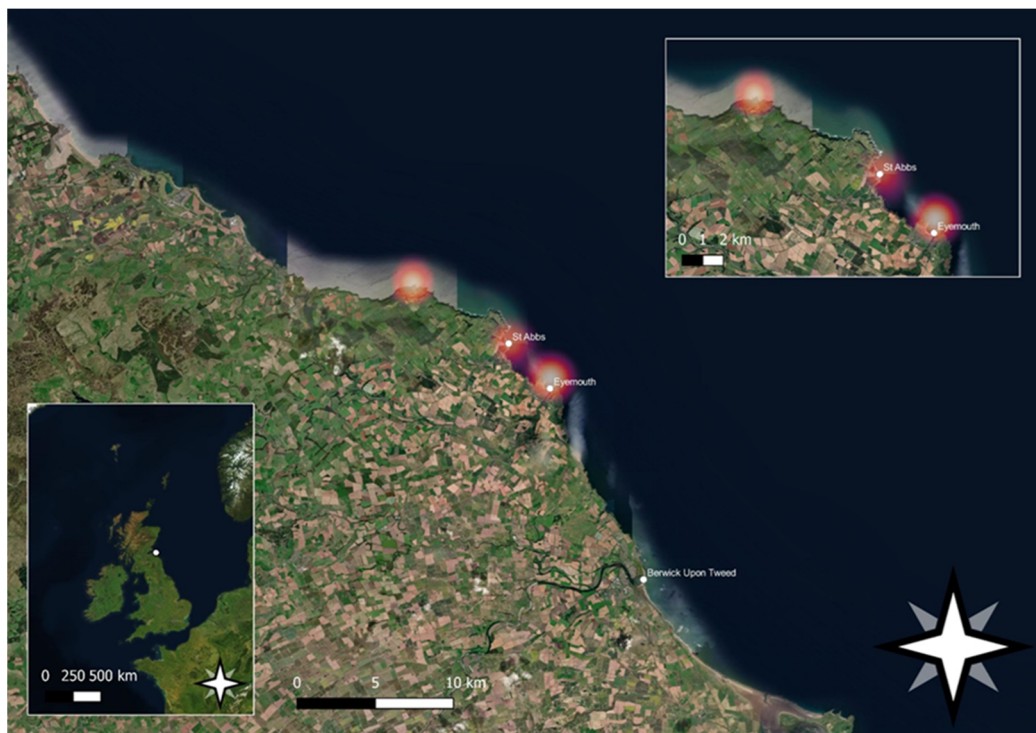

**Figure 8.** Heat map of the gonad maturity stages for both male (*n* = 313) and female (*n* = 124) *C. pagurus* individuals sampled in Berwickshire (55.902769, −2.128988). Gonad maturity stages three and four were considered as they correspond to sexual maturity for both sexes. The white colour of the map highlights the later maturity stage; in this case, gonad stage four, with gonad stage 3 emitting the pink/purple colour.

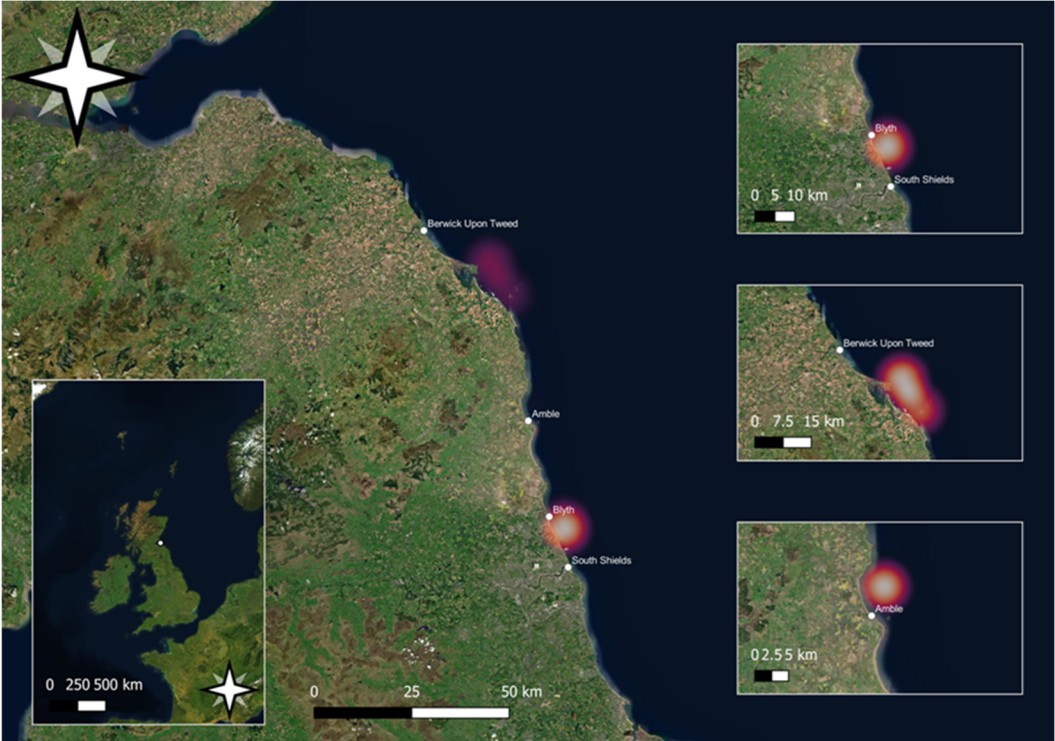

**Figure 9.** Heat map of the gonad maturity stages for both male (*n* = 188) and female (*n* = 144) *C. pagurus* individuals sampled in Northumberland (55.137154, −1.437651). Gonad maturity stages three and four were considered as they correspond to sexual maturity for both sexes. The white colour of the map highlights the later maturity stage; in this case, gonad stage four, with gonad stage three emitting the pink/purple colour.



*4.2. Morphometric Maturity*

Using morphometric measurements to determine onset of sexual maturity is commonplace in fisheries research [7,8,21,26,30,31,33]. When discussing the size at sexual maturity, morphometric data must be considered as an estimate due to the regional variations in individuals' growth rate and age at maturation [26]. The typical metric for males is the chelae and the abdominal flap widths for females as they indicate sexual dimorphism [7,26]; however, in the present study the same metrics were used across both sexes, the carapace width and chelae height, which followed the protocol of the sizeMat package. Behaviours indicative of courtship and combat signify a change in the allometry of male chelae, whereas a change in abdominal width for females relates to the accommodation of egg clutches [27]. It may be suggested that the nuance of sexual dimorphism in the females sampled may be lost due to the lack of abdominal flap measurements used in the analysis. The onset of sexual maturity based on the morphometrics on chelae length and carapace width was assessed by [31], whereby the allometry was met at 107 mm for males and 155 mm for females. Considering the results from the segmented regressions, males in Berwickshire (119.06 mm) and Northumberland (129 mm) show morphometric maturity at much greater sizes. Using morphometric measurements as the sole method of assessing maturity in this species provides inadequate results, as mentioned in [7,8], as mature males are underestimated, and mature females are overestimated. In this present study, the difference in the morphometric and gonadal maturity estimates varied. A difference of 32.6 mm for males, 7.9 mm for females in Berwickshire and a difference of 20.2 mm for males in Northumberland was found. Only the females from the Northumberland cohort showed no variation in estimates between the morphometric and gonadal $CW_{50}$ values. A sex difference in maturation was stated by [15,26,27] in that males mature at smaller sizes than females which was stated to be documented by [10] also. This pattern of smaller maturation sizes in males compared to females was not found in both regions. A carapace width size difference of 6.4 mm in Berwickshire and 9.3 mm in Northumberland was observed, if morphometric maturity is considered solely. Sample size for the present study was much greater than that of [26], which could contribute to the contrast in patterns regarding smaller sizes of maturation in males. Fecundity of *C.pagurus* significantly increases with the size of the female carapace [14]. It has been recorded that age of maturity for female *C. pagurus* is four years which was the oldest age of maturation recorded by [14], and the reproductive cycle was predicted to be seven years which included an annual or two annual broods [14]. Fishing pressure is documented to change growth patterns over ecological and evolutionary time periods [7,28]. In this instance, ecological change by fishing pressure may suggest the change in SOM for the Northumberland cohort when NIFCA historic data in [33] are compared with this study. The NIFCA Byelaw: "Crustacea and Molluscs Permitting and Pot Limitation" states commercial fishermen in the district are limited to 800 fishing pots, whereas in the Berwickshire region there is no pot limitation. The fishing pressure in each region greatly varies and therefore regional variances in the size at onset of maturity would be expected, considering the suggestion by [7]. By applying movement tracking to the wider *C. pagurus* population, the ability to highlight whether these populations are indicatively different would show whether these populations are subject to variations in fishing pressure as we cannot suggest whether mature individuals migrate to and from these regions and thus affect the general SOM in each respective region.

*4.3. Hepatopancreas Weight and Hepatopancreas Index (HSI)*

It has been suggested that the HSI values in edible crab follow a capitalist strategy [7,21]. This suggests that the energy used in reproduction is not recovered post mating. HSI is a means of understanding the stored fats in the individual [7,21], as well as the reproductive cycle of gamete production in both sexes [7,16,21,26]. In this study, it could

be perceived that this trend of energy loss during reproduction and lack of recovery post mating is found in the sampled population from Berwickshire but not for those from Northumberland (Figure 6). Not only from reproduction, but other factors could suggest the variations in HSI values such as the redirection of energy to chelae and limb regrowth and the negative effects of black spot disease whereby an energy deficiency is met when the health of the individual worsens, influencing the individual to divert energy stores towards reproduction [37,17]. Further investigation into factors, such as sea temperature and diseases, that could influence the HSI values of this species is required.

*4.4. Gonad Weight and Gonadosomatic Index (GSI)*

The GSI is determined as the means of measuring the reproductive timing of the individual as well as the spawning season for a group [39]. It was found that the GSI was significant to sex, gonad stage, and condition of the individual (Table 8). Similar results were found in [7]. Only in Berwickshire, the GSI was not significant with condition. The GSI generally increases with gonad stages following the maturation cycle of the individuals [40]. This has been observed by [16], and also in the present study (Table 8). GSI has also been shown to increase with black spot disease and in individuals considered to be in poor health [17,18,26], suggesting that in times of physical stress the individual will redirect energy into reproduction but there is no evidence of such behaviour.

**5. Conclusions**

This study highlights the variations between regions regarding the size at which sexual maturity is reached in the commercially important *C. pagurus*. The SOM (108.5–126.8 mm) found in this study allows for the maturing population to reproduce at least once before they are fished from this stock when related to the specific MCRS standards in both regions (150 mm Berwickshire and 130 mm Northumberland). It would be suggested then that the current MCRS values are effective at maintaining a viable brood stock for both regions. However, the variation in these regions still warrants the need for localised and regional monitoring of this species. A consistent monitoring procedure would be of benefit as anecdotal evidence suggests that fishermen are landing the 130 mm-sized individuals in Northumberland to counter the 150 mm legal limit in Berwickshire. NIFCA, through their monitoring activities, have a greater understanding of their fishery than would be stated for Berwickshire. Monitoring in Scotland is conducted over a larger scale which limits the nuances that are apparent in smaller, more localised, marine areas and therefore this study provides information that is specific and not generalised to Berwickshire. Further investigation into cross landing of sized individuals is required as removing individuals from the population that are not meeting the regional SOM may lead to a reduction in the stock. The data collected in this study provide a baseline for the region of Berwickshire, whilst providing additional information to the Northumberland Inshore Fisheries and Conservation Authority (NIFCA).

**Author Contributions:** Conceptualization, B.A.A.E., A.B. and J.R.; methodology, B.A.A.E., A.B. and J.R.; validation, B.A.A.E., A.B., J.R. and K.S.; formal analysis, B.A.A.E.; investigation, B.A.A.E., A.B. and J.R.; resources, B.A.A.E., A.B., J.R. and K.S.; data curation, B.A.A.E.; writing—original draft preparation, B.A.A.E.; writing—review and editing, B.A.A.E., A.B., J.R. and K.S.; visualization, B.A.A.E.; supervision, B.A.A.E.; project administration, A.B. and J.R.; funding acquisition, A.B. and J.R. All authors have read and agreed to the published version of the manuscript.

**Funding:** This research was supported through monetary contributions from Northumberland Inshore Fisheries Conservation Authority (NIFCA), Blue Marine Foundation and also in-kind monetary contributions by the Nesbitt-Cleland Trust.

**Institutional Review Board Statement:** The research adhered to the legal requirements of the country (U.K.) in which the work was carried out, and all institutional guidelines of St Abbs Marine Station. Informal ethical review and approval of this research was conducted by authors and trustees of St Abbs Marine Station using published literature and information obtained from

Animals (Scientific Procedures) Act 1986 (ASPA). Animal collection was conducted with the local commercial fishing fleet. Derogations were granted by Marine Scotland over the sampling period to allow the landing of individuals below the Scottish MCRS of 150 mm. The surveys were in accordance to terms of Section 9 of the Sea Fish Conservation Act 1967, Article 25 of Council Regulation No. 2019/1241, the specified crustaceans (prohibition on landing, sale and carriage) (Scotland): order 2017 No.455 and the undersized edible crabs (Scotland) order 200 No.228. At present crustaceans and cephalopods still remain outside the protections of the Animal Welfare Act, the Act which makes it an offence to cause unnecessary suffering to protected animals. The experiment does not need ethical approval.

**Data Availability Statement:** The data collected and analysed in this paper is not available publicly. Readers can inquire and request the raw data files from the first author (blair.easton@marinestation.co.uk).

**Acknowledgments:** We would like to thank the local inshore fishermen from St. Abbs and Eyemouth and also the efforts of our colleagues at NIFCA for providing the necessary number of *C. pagurus* for this study to be conducted. We would also like to thank our colleagues at St. Abbs Marine Station, NIFCA and Blue Marine Foundation for their help through the collection, dissection phases of this project, and their constructive input during the writing phases.

**Conflicts of Interest:** The authors declare no conflict of interest. The funders had no role in the design of the study; in the collection, analyses, or interpretation of data; in the writing of the manuscript; or in the decision to publish the results.

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
