# Peer review of "Comparing the Size at Onset of Sexual Maturity of Edible Crab (Cancer pagurus, Cancridae) in Berwickshire and Northumberland"

_fishes, doi:10.3390/fishes8050260_

Round 1
Reviewer 1 Report
Table 2: Please double check numbers of references. They may be wrong.
Line 103: Could you please add the sampling period? When did you collect data?
Line 127: Reference number 11 does not include information about the blackspot. Please check reference numbers throughout the text.
Lines from 569 to 662: Please check references. Some papers were written two or more times.
Author Response
Dear reviewer,
Thank you for your comments and your time for reading the manuscript.
I have attached the edited draft of the manuscript taking into account your comments and the second reviewers. I believed this would be more appropriate for reviewing the edits and amendments.
Regarding your comments:
References in table 2 have been checked and amended.
The sampling period has been added to Line 103.
Reference 11 on line 127 has been amended.
The reference list has been checked and references which had been repeated have been removed.
Thank you again for your time and feedback on the initial manuscript review.

Reviewer 2 Report
Dear authors,
The manuscript is relevant but I think to adjust some informations. To improve this study I left some suggestions in the text.

Author Response
Dear reviewer,
Thank you for your comments and the time you gave to review the manuscript.
I have attached the edited and amended manuscript based on your comments and the other reviewer. I believed this would be more appropriate for reviewing said amendments.
Regarding your comments:
Line 151 - I did not take images/photos of the dissected individuals before gonad removal. Therefore, I do not have images of the hepatopancreas.
Line 195 - you stated the histological analysis as a basis to negate the potential for young individuals who are mature from providing a false positive. The analysis in this section is based on the histological analysis of the gonads in relation to the carapace width to prevent such false positives from influencing the statistical results.
Line 273 - I did not test for morphometric maturity changes over the sampling period. The results are a collective of all sampled individuals across the full sampling period.
Line 395 - I have amended the sentence and added a reference relating to chemical effects on the maturation of the species.
Figures have been amended as per your comments.
Thank you again for your comments and your time for reviewing the manuscript.

Round 2
Reviewer 1 Report
Dear authors,
Please check references again. There is still mistake related to cited paper. In table 2, reference number 30 should be changed with reference number 43. In other words, SOM values from Isle of Man were reported by Öndes et al. 2017. Thus, aforementioned paper (43) should be cited in the table 2.
Congratulations.
Author Response
Dear Author,
Thank you for your time and comments on your second review of the manuscript.
I have attached the mauscript with the references amended based on your comments. The reference in Table 2 has been changed to reference number 43. Other amendments have been made due to other reference numbers appearing one number higher than they should.
Thank you again for your time.

Reviewer 2 Report
Dear Author,
Congratulations to add informations in the manuscript. The text is very important to others searching about this subject.
Author Response
Dear Author,
Thank you for your time and comments on your second review of the manuscript.
I have attached the new reviewed manuscript as the other reviewers noticed reference numbers in the wrong location. These have been amended per their comments.
Thank you again for your time.

Round 3
Reviewer 1 Report
Thank you.